# Vitamin D—A Risk Factor for Bone Fractures in Children: A Population-Based Prospective Case–Control Randomized Cross-Sectional Study

**DOI:** 10.3390/ijerph20043300

**Published:** 2023-02-13

**Authors:** Alexandru Herdea, Adelina Ionescu, Mihai-Codrut Dragomirescu, Alexandru Ulici

**Affiliations:** 111th Department of Pediatric Orthopedics, “Carol Davila” University of Medicine and Pharmacy, Bd. Eroii Sanitari nr. 8, 050474 Bucharest, Romania; 2Pediatric Orthopedics Department, “Grigore Alexandrescu” Children’s Emergency Hospital, 011743 Bucharest, Romania

**Keywords:** vitamin D, deficiency, children, fracture risk, fracture healing

## Abstract

Background: Vitamin D is an essential component in calcium metabolism. Seasonality, advanced age, sex, dark skin pigmentation, and limited exposure to sunlight were reported as causes of vitamin D deficiency. This study aims to determine whether children with lower levels of vitamin D suffer more fractures than those with sufficient levels. Materials and Methods: Our institution underwent a prospective case–control randomized cross-sectional single-blinded study that included 688 children. They were split into two groups: the study group and the control group. The study group received supplements of vitamin D and calcium for 6 months. Another reference cohort was observed, which comprised 889 patients in the pediatric ward for different respiratory or gastroenterological conditions without a history of fractures. This group was used for age–sex matching tests. Results: Logistic regression showed that with every one unit increase of vitamin D level, the chance of having a middle third fracture in both bones of the forearm decreased by 7% (OR 1.07); distal third fracture incidence decreased by 1.03 times; middle third radius fracture incidence decreased by 1.03 times; distal third radius fracture incidence decreased by 1.06 times. The risk of having a distal third both-bone forearm fracture increased by 1.06 times with every year of age. Comparing the healing process, we noticed an improvement in bony callus formation for patients in the study group. Conclusions: Dosing the serum level of 25-OH-vitamin D should be taken into consideration for pediatric low-energy trauma fractures. Supplementing with vitamin D and calcium throughout childhood can be a solution for healthy bones. Our preliminary results show that the normal level of vitamin D in children should start at 40 ng/mL.

## 1. Introduction

The role of vitamin D expands beyond that of bone mineralization. It is known to be involved in a plethora of biological processes, including neoplasia and autoimmune diseases, viral infections, and musculoskeletal illnesses. For the purpose of this study, we will focus on the relationship between the level of vitamin D and bone metabolism by assessing a group of children who suffered from traumatic injuries to the skeletal system during a predetermined period.

Vitamin D, along with parathyroid hormone (PTH) and calcitonin (CT), are the three principal effectors of calcium and phosphorus homeostasis [1]. Together, they sustain bone health and osteogenesis. A sufficient level of 25-hydroxy-vitamin D for adults is 30 ng/mL [2]. Any level under this value is classified either as insufficiency (20–30 ng/mL) or deficiency (under 20 ng/mL) [3]. For children, a serum range of 40–70 ng/mL is recommended for optimal health benefits and to reduce the health costs of a series of diseases influenced by vitamin D, such as influenza, asthma, upper respiratory infections, and cancer [4]. In cases of increased bone fragility, the recommended level should be at least 30 ng/mL [4].

It is estimated that vitamin D deficiency affects one billion people worldwide across all ethnicities and age groups [5]. Vitamin D insufficiency affects almost 50% of the general population [5,6]. One study showed that in Romania, only 40% of the population had sufficient vitamin D [7]. It is an endemic area for rickets, and the prophylaxis includes daily vitamin D intake until the age of 24 months, then once a day for “months with reduced light exposure (September-April)” until the age of seven [8]. It is also known that the vitamin D necessity is high during puberty, but there is a lack of an established protocol for this age category [9]. Further supplements throughout childhood may be necessary [10].

In the adult population, there is evidence linking vitamin D deficiency, along with low bone mineral density, to fracture risk [11]. In children, there is no predictive algorithm to determine fracture risk [12]. It is unclear if there is a link between low vitamin D and why certain children sustain fractures from minor trauma whereas others do not. Further studies are needed to confirm the link between vitamin D levels and fractures, bone healing, and risk of refracture. Our manuscript aims to evaluate correlations between these individual variables.

Winter and spring seasons, older age, female sex, dark skin pigmentation, and limited exposure to sunlight were reported as causes of vitamin D deficiency [13,14]. The US, Canadian, and European reference values used for vitamin D are between 400–800 IU/day [15]. However, daily vitamin D intake varies widely depending on regional customs, supplementation, and the availability of fortified food items. Patients with bone or kidney disease are defined as a vulnerable population and may require between 600–1000 IU/day [2]. It is to be determined in further studies whether the current necessity should be the same for all children or if it should be tailored based on physical and environmental characteristics. Low vitamin D and calcium status cause secondary hyperparathyroidism, which promotes bone demineralization and increased fracture risk [15]. A significant association between calcium intake and fracture risk was assessed by Alshamrani et al. [16], but no association between vitamin D and fracture risk was found.

Pathological conditions such as nonspecific symmetrical non-radiating muscle pain, pathological bone fractures, slipped capital femoral epiphysis, back pain, muscle cramps, and pain during running may suggest insufficient or low vitamin D levels [17]. Vitamin D was also seen as a risk factor for curvature progression among children with scoliosis [18,19]. According to Perez-Rossello, bone demineralization is associated with a median value of 25-OH-vitamin D of 7 ng/mL [20].

The Global Consensus Recommendations on Prevention and Management of Nutritional Rickets state that children with radiographically confirmed rickets have an increased risk of fracture [21].

We hypothesized that children with lower levels of vitamin D suffer more fractures and that their healing process is delayed compared to children with normal levels of vitamin D suffering the same type of fracture.

## 2. Materials and Methods

### 2.1. Study Design

The study took place in the Pediatric Orthopedics Department of “Grigore Alexandrescu” Children’s Emergency Clinical Hospital, located in an urban area in Bucharest, Romania. The ethics committee of “Grigore Alexandrescu” Children’s Emergency Clinical Hospital approved this study on 27 February 2018, and the identification number of the survey is 24. Consent from the parents of all participating patients was obtained. The study took place between 2018–2020. A patient flow diagram is shown in Figure 1.

### 2.2. Participants

A prospective case-control randomized cross-sectional single-blinded study was conducted between 2018 and 2020. The study included children referred to the clinic for various fractures of the upper and lower extremities. After fracture diagnosis, they were split into two groups: the study group, which included patients that would receive calcium and vitamin D supplements, and the control group, which included patients that would not receive supplements. Another reference cohort comprised patients presented in the pediatric ward for respiratory or gastroenterological conditions without a history of fractures or intake of vitamin D and calcium in the last 6 months. This group was used for age–sex matching tests. Vitamin D and calcium levels were available for all of the patients included in the study. Matching was done for age, gender, and season.

Randomization was done with the electronically stored medical registration program by automatically including a patient with a fracture randomly in either the study or the control group; thus, the risk of bias was kept to a minimum (none of the parties knew which group a patient would enter).

The working protocol for the patients in the study and control group included the following aspects: patient history, clinical and radiological exam, type of fracture, blood samples for 25-OH-vitamin D and calcium, and follow-up to see the time of healing and fracture recurrence.

Patients’ electronic files were classified according to ICD-10 (10th revision of the International Statistical Classification of Diseases and Related Health Problems). Injuries were defined as all trauma presentations with an ICD-10 code in the categories beginning with the letter ‘S’ and then supplemented by the causal code (how the fracture occurred). Four authors examined all data and corrected inconsistencies in the database to reduce errors, such as diagnostic equivalents (i.e., distal radial fracture vs. distal radial epiphyseal separation vs. distal radial Salter–Harris 2 fracture).

X-rays were taken on 2 sides (for example, forearm AP and lateral, hand AP, and ¾ or oblique view) and were digitally stored using a high-precision digital system.

Inclusion criteria for the study and control group were as follows: age between 2–17 years, fractures of upper and lower limbs, clinical and radiological exam to sustain the diagnosis, follow-up for at least 12 months, 25-OH-vitamin D and calcium blood samples at the time of enrollment in the study and after 6 months of supplementation (only for the study group).

Exclusion criteria included age less than 2 years, age above 18 years, abusive trauma (including Silverman syndrome), high-energy trauma mechanisms, child abuse cases, pelvic and spine fractures, patients suffering from chronic diseases affecting bone status (including osteogenesis imperfecta), dysplasia, cystic fibrosis, patients treated with corticosteroids, follow-up for less than 12 months, lack of patient history, missing blood samples, and intake of vitamin D and calcium treatment in the 6 months prior to the study.

### 2.3. Study Procedure

After the randomization procedure, patients were allocated to either the study or control group. Both groups were required to take blood samples of 25-OH-vitamin D and calcium, and they followed through with the standard treatment procedure. The standard treatment procedure was based on fracture complexity but included casting, closed reduction, and open reduction. X-ray and clinical follow-ups were done after 1 day, 3 weeks, 5 weeks, 2 months, 6 months, and 1 year.

Patients in the study group were asked to perform the following blood tests: 25-OH-vitamin D and total calcium. They repeated them after 6 months of supplementation.

For vitamin D, the values for 25-OH-vitamin D were classified with standard laboratory values as follows: toxic value: over 100 ng/mL, normal value: between 30 ng/mL and 100 ng/mL, poor value: between 20 ng/mL and 29 ng/mL, and insufficient value: less than 20 ng/mL. For calcium, the values of total calcium were considered as follows: hypercalcemia: over 10.6 mg/dL; normal value: between 8.80 mg/dL and 10.6 mg/dL.

The following therapeutic interventions were applied to the study group for the following 6 months if vitamin D was less than 30 ng/mL: oral treatment: patients would receive vitamin D [2000 I.U./day] and calcium [600 mg/day] in the first part of the day.

The control group performed blood tests at the beginning of the study and was monitored clinically and radiologically for fracture healing and functional outcomes.

The study duration was at least one year for each patient included.

### 2.4. Statistical Analysis

Data was collected using the institutional informatic system. Data included categorical qualitative data (gender, type of fracture, and season) and continuous quantitative data (age at time of diagnosis, and levels of vitamin D and calcium). Descriptive tests such as frequencies, sex incidence, odds ratio, chi-square tests, and *p*-value were conducted. A result was considered statistically significant if the *p*-value was less than 0.05. The statistical analysis was performed by using RStudio version 1.1.447 R 3.6.0 (Posit Software, PBC formerly RStudio, PBC, Vienna, Austria) and Microsoft Excel Office 2016 (Microsoft, Redmond, Washington, DC, USA). We used a 95% confidence interval.

## 3. Results

One thousand, five hundred and seventy-seven patients were evaluated in the study. In the selected timeframe, the study included 688 children with fracture diagnoses that met the inclusion criteria. After the randomization method, they were split into two groups: the study group, which included 357 patients, and the control group, which included 331 patients. The reference cohort (no history of fractures) included 889 patients.

The control group was composed of 162 (48.9%) male patients and 169 (51.05%) female patients, whereas the study group consisted of 237 (66.8%) boys and 120 girls (33.6%). The reference cohort included 439 male patients (49.8%) and 450 female patients (50.61%). The distribution of patients into cohorts based on gender and age is shown in Table 1.

After evaluating the levels of vitamin D and calcium for each patient, we split them into two major groups as follows: the fractured group, which included patients from both control and study cohorts, and the unfractured group, which included all of the patients from the reference cohort (Table 2).

The table exposes the difference between the mean levels of vitamin D in the fractured group (25.29 ng/mL) and the unfractured group (30.99 ng/mL), which is almost 5 ng/mL. It can also be noticed that in the fractured group, girls had slightly higher levels of vitamin D than boys, whereas in the unfractured group, boys had higher levels.

The mean value of the vitamin D levels in the study group was 23.91 ng/mL, and it was 26.73 ng/mL in the control group (*p*-value = 0.000412). The result is statistically significant, but both groups can be included in the deficient level group, and the results can be considered similar. As for calcium mean levels, the study group registered 9.92 mg/dL versus 9.98 mg/dL in the control group (*p*-value = 0.211).

Out of 688 patients with fractures, 683 had upper limb fractures, and 5 had lower limb fractures.

Table 3 shows the mean levels of vitamin D and calcium in the patient study and control group according to gender and frequency per fracture type. Only the first six types were selected according to the total number of patients included.

The mean values of vitamin D levels for both genders were compared (Table 4). A statistically significant difference between male and female patients’ levels of vitamin D3 was observed (*p* = 4.15 × 10^−8^ and 0.0003 in the study group and control group, respectively) in favor of the female population (22.33/24.93 ng/mL in male patients in the study/control group vs. 27.03/28.51 ng/mL in female patients in the study/control group).

Using a paired *t*-test for means (Table 4) we found a statistically significant difference between the initial level of vitamin D and after 6 months of treatment with vitamin D supplements in the study group (*p* = 1.20 × 10^−31^). No statistically significant difference was revealed between the initial measurement of calcium levels and after 6 months of supplementation with calcium (*p* = 0.13).

Results showed that with every one unit increase in vitamin D level, the chance of having a middle third fracture in both bones of the forearm decreased by 7% (OR 1.07).

One hundred and forty-eight patients suffered distal third fractures of both forearm bones. The risk of having this type of fracture increased by 1.06 times with every year of age. Female patients presented 2.63 times fewer chances to suffer this type of fracture than boys. The chance of having this fracture decreased by 1.03 times with each unit increase of vitamin D serum level.

One hundred and twenty children suffered a middle-third radius fracture. Age, sex, and level of vitamin D matching tests revealed that by every one-year increase of age, the risk of having a distal third radius fracture decreased 1.05-fold. Female gender increased the chances of this type of fracture occurrence by 2.88 times. For every additional unit of vitamin D serum level, the risk of suffering this fracture was reduced by 1.03 times.

One hundred and sixteen children presented a middle-third fracture of both bones of the forearm. Female patients had a 3.44 times lower chance to suffer this type of fracture than male patients. The chances of presenting this fracture decreased 1.03-fold with every one-unit increase in vitamin D serum level.

The distal third radius fracture subgroup included 116 subjects. Each increment in vitamin D serum levels decreased the chances of presenting this type of fracture by 1.06.

By using the logistic regression model on the fractured groups, the 25-OH-vitamin D level was found to be a protective factor starting from 40 ng/mL and above.

The entire observed fractured population (1577 patients) was age–sex matched. They were grouped by age and sex to evaluate the effect of vitamin D serum level on fracture occurrence. Vitamin D levels were considered either normal (>30 ng/mL) or under normal (<30 ng/mL) (Table 5).

We ran a conditional logistic regression model (*p* = 2.11 × 10^−9^). Regardless of age and sex, we concluded that patients having an under-normal serum level of vitamin D had a 2.127 times higher chance of having a fracture.

A total of 1577 children were included in the study and control groups. 517 children with fractures had under-normal levels of vitamin D serum levels, whereas only 171 patients presented normal vitamin D levels. In the unfractured group, 494 children had under-normal serum levels of vitamin D, and 395 subjects had normal serum levels.

There were statistically significant differences between vitamin D serum levels in the two populations in the following age and gender groups: 4–6, 7–10, 11–14, and 15–17 years old male patients and 2–3 and 4–6 years old female patients (*p* < 0.05).

Considering the four seasons of the year, seasonal variation of vitamin D was tested. Patients had lower serum levels of vitamin D if their testing had been run in spring or winter (24.78 ng/mL and 27.57 ng/mL, respectively), and they had higher levels in autumn or summer (30.15 ng/mL and 29.33 ng/mL, respectively).

Comparing healing processes, we noticed an improvement in bony callus formation for patients in the study group, as seen in Figure 2. We had the patients compared at the same time in healing evolution for 6 months or more with patients of the same fracture type the same lateral and AP x-rays, and those that were age–sex matched. We noticed that bony callus was even more present for the patients that received vitamin D and calcium in each healing phase. No adverse effects were noted after supplementation with vitamin D and calcium for at least 6 months. No long-term complications of a fracture were noted in our study population. All patients healed and resumed activities.

## 4. Discussion

One in two children between 5 and 18 years of age will suffer at least one fracture, according to a study conducted by Jones IE on a patient pool of 601 [22]. In agreement with the results from an 11-year study in Sweden, at least one-third of children under 17 years will suffer one or more fractures [23]. There is a series of studies in pediatric orthopedics that suggests a link between the plasma level of vitamin D and the occurrence of fractures [24,25,26,27,28].

In our study we followed 688 patients between 2 and 17 years of age that suffered a fracture and 889 patients that had no history of fractures, and it was supplemented with a follow-up period of at least 1 year.

Vitamin D is a topic of public health interest. Besides its role in calcium homeostasis, it is a multifunctional substance newly classified as a hormone. The optimal serum level in children is yet to be clarified, despite much published research, as well as the optimal protocol for children. Some authors considered sun exposure to be the only healthy way of acquiring vitamin D, whereas others found refractory rickets in the tropics [29].

According to Endocrinology Society guidelines, a serum level of 30 ng/mL (75 nmol/L) is considered normal. A level of under 30 ng/mL but over 20 ng/mL is considered insufficient, and less than 20 ng/mL means deficiency [30].

From our bone fracture study group, we found 31.39% of the patients had insufficient vitamin D levels, 43.75% with deficient levels, and only 24.85% with normal levels. In the unfractured control group, 44.43% of the patients had normal serum vitamin D levels.

Some authors suggest reconsidering these target levels based on age groups and medical conditions [28]. Some experts consider that patients with bone fragility should have a higher level of D3 and should be treated with an initial higher dose followed by maintenance.

In order to increase the serum levels for the bone fracture patients, 2000 I.U./day of vitamin D and 600 mg/day of calcium were given as oral treatment for the following 6 months. We found an average increase of 53.44% in their vitamin D serum levels (from 23.9 ng/mL to 44.76 ng/mL, *p* = 1.20 × 10^−31^) and an increase in their calcium serum levels from 9.98 mg to 10.05 mg, which is not statistically significant (*p* = 0.13).

The overweight/obese patient is associated with a lower serum level of vitamin D. Other signs that may suggest an insufficient level of D3 are atypical muscle pain, pathological bone fractures, epiphysiolysis, weight-bearing joint pain, back pain, difficulty in climbing stairs, pain during running, and muscle cramps. In addition, this pain is symmetrical, without radiation, and accompanied by local tenderness [17].

Kwon DH et al. conducted a study on 163 children who presented a fracture and found that 80% of them had vitamin D deficiencies [31]. The authors suggested that patients from urban areas should be monitored regularly for their serum levels [31]. In our study, 75.14% of the bone fracture population had below normal vitamin D serum levels. Monitoring the unfractured bone cohort, we noticed that only 55.57% of them were below normal vitamin D serum levels.

Thompson et al. concluded in their study that higher fracture incidence is associated with vitamin D insufficiency in the pediatric population [27]. From our results, we observed a 2.127 times higher risk of having a fracture among children with under-normal vitamin D serum levels, regardless of age or sex.

We found a mean difference of 5 ng/mL between the fractured and unfractured bone groups. Hence, we found a relationship between fracture risk and vitamin D levels among children. We concluded that patients having an under-normal serum level of vitamin D had a 2.127 times higher chance of having a fracture than children with normal levels.

According to a study made by Scheimberg [32], no black or Asian children had normal levels of vitamin D, whereas only 5% of white children had optimal levels. Other studies show that Asia, Africa, and the Middle East have high incidences of rickets/osteomalacia with low levels of vitamin D [17]. A retrospective study conducted on 234 children from Saudi Arabia with a history of fractures found significantly lower levels of vitamin D than the control group. They suggested supplementation with vitamin D for the general population [26]. Mansour MM et al. measured serum levels of D3 on 510 children admitted at Jeddah hospital, Saudi Arabia for varied reasons other than fractures, and they observed that 79% were in deficit [33]. Moreover, 27% of them had serum levels under 7 ng/mL. The authors suggested not only vitamin D administration but also fortification of food with vitamin D for the pediatric population [34]. In Egypt, El-Sakka A et al. measured serum levels among children with forearm fractures and noted low bone mineral density along with vitamin D deficits [33].

A study made in 2002 by Jones et al. observed a group of 601 children who were monitored between the ages of 5 and 18 and discovered that one in two children suffered at least one fracture during childhood [22]. Gorter found more cases of calcidiol deficiency in older patients (over 10 years) [35]. He demonstrated that higher age, non-Caucasian ethnicity, and winter and spring seasons are risk/protective factors for vitamin D deficiency. According to this study, 74% of children with low levels of vitamin D increased their values after four months of treatment. However, children living in developed countries have lower than half of the recommended intake [28]. The consequences of the low intake and below-target levels of vitamin D and calcium are yet to be established because of little research using case–control studies.

The Global Consensus Recommendations on Prevention and Management of Nutritional Rickets state that screening for vitamin D deficiency by obtaining a serum 25-OH-D concentration is recommended in patients at increased risk of bone fragility and in those with recurrent low-impact fractures. For these groups, a serum level of 30 ng/mL is considered adequate [21].

Minkowitz et al. had a patient pool of 1031 patients and noted that children with lower levels of vitamin D suffered more severe fractures. They suggested a target level of 40 ng/mL, which is in agreement with our preliminary results [36].

A single case report suggests that hypovitaminosis D is a possible cause of inadequate fracture healing and refracture in children, and it shows a clear effect of vitamin D supplementation on callus formation in the absence of additional variables [37]. In our study, our findings also show that bony callus formation was seen sooner among those who received vitamin D and calcium supplementation. More studies are needed to correlate timelines for bony callus formation with the intake of vitamin D, calcium, and other factors that may contribute.

Our study’s strength was an adequate patient pool with sufficient data to sustain a thorough statistical analysis. One of our study’s limitations was the lack of long-term follow-up. Future research is needed to see if formerly deficient children who received supplements and have a normal level of vitamin D have the same fracture incidence as the general population.

## 5. Conclusions

The involvement of vitamin D in the occurrence and consolidation of fractures in the pediatric population may play a role in those fractures; therefore, we recommend dosing the serum level of 25-OH-vitamin D, especially in the case of low-energy trauma fractures.

Supplementing with vitamin D and calcium throughout childhood can be a solution for a healthy skeletal system, and in case of a fracture, it can help the consolidation process.

Our preliminary results show that the correct level of vitamin D to prevent some fractures in children should start at 40 ng/mL.

## Figures and Tables

**Figure 1 ijerph-20-03300-f001:**
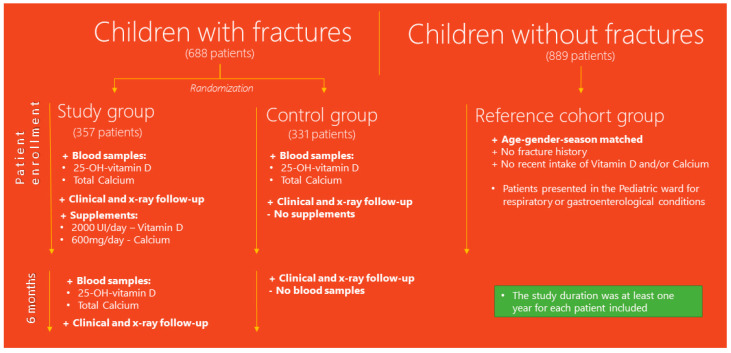
Flow diagram describing the patient selection process of the study group, control group, and reference cohort. The study duration was at least one year for each patient included.

**Figure 2 ijerph-20-03300-f002:**
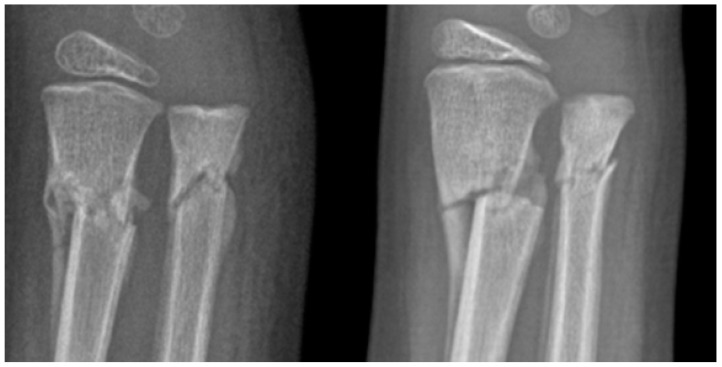
Image presenting a 1/3 distal both-bone forearm fracture of two 6-year-old patients that underwent close reduction. X-ray evaluation at 3 weeks is shown. **Left image**—6-year-old—control group (no supplements)—27 ng/mL vitamin D. **Right image**—6-year-old—study group (supplements)—26 ng/mL vitamin D. More bony callus can be seen on the **right image**.

**Table 1 ijerph-20-03300-t001:** Patient distribution according to age group, gender, and group type.

Age Group:	2–3 Years		4–7 Years		8–10 Years		11–14 Years		15–17 Years		Total
Gender:	M	F	M	F	M	F	M	F	M	F	
Control group	0	1	34	104	21	56	101	8	6	0	331
Study group	1	28	45	39	46	46	135	7	10	0	357
Reference group	103	84	95	103	105	98	97	107	39	58	889
Total	104	113	174	246	172	200	333	122	55	58	1577

**Table 2 ijerph-20-03300-t002:** Patient distribution according to age group, gender, fractured/unfractured group, and 25-OH-vitamin D level in ng/mL.

Age Group (Years):	2–3		4–6		7–10		11–14		15–17		Total
Gender:	M	F	M	F	M	F	M	F	M	F	
Fractured group	1	29	79	143	67	102	236	15	16	0	688
Vitamin D level (ng/mL)	29.00	32.38	25.65	28.49	26.57	27.45	21.80	21.88	21.10	-	25.29
Unfractured group	103	84	95	103	105	98	97	107	39	58	889
Vitamin D level (ng/mL)	38.05	38.45	34.43	33.24	30.31	27.58	27.66	24.86	26.83	24.62	30.99
Total	104	113	174	246	172	200	333	122	55	58	1577
Vitamin D level (ng/mL)	37.97	36.89	30.96	31.35	28.65	27.50	23.51	24.50	25.16	24.62	28.50

**Table 3 ijerph-20-03300-t003:** Patient distribution according to the most frequent fracture type encountered, study groups, and the mean levels of vitamin D and calcium. Only the first six types of fractures based on the total number of occurrences are shown.

	Control Group			Study Group		
Fracture Type	Total	Mean. Vit. D3 (ng/mL)	Mean Calcium (mg/dL)	Total	Mean. Vit. D3 (ng/mL)	Mean Calcium (mg/dL)
1/3 distal of both forearm bones	76	26.76	9.99	76	24.8	10.3
1/3 middle radius	75	17.9	9.8	76	24.14	9.8
Buckle fracture of the radius	44	21.35	9.86	72	23.92	9.93
1/3 distal radius	40	13.3	9.75	54	17.9	10.2
1/3 middle of both bones forearm	40	24.23	9.87	45	24.5	9.67
Buckle fracture of both bones of the forearm	27	25.8	9.8	10	24.57	9.16

**Table 4 ijerph-20-03300-t004:** Comparison of vitamin D and calcium blood levels between enrollment in the study and after taking the supplements for 6 months.

Blood Levels	Date	Mean	Dispersion	*p*-Value
Vitamin D	Initial	23.92	46.46	1.20 × 10^−31^
	After 6 months	44.76	21.55	
Calcium (mg/dL)	Initial	9.98	0.13	0.13
	After 6 months	10.05	0.04	

**Table 5 ijerph-20-03300-t005:** Patient distribution according to fractured or unfractured group and, subsequently, according to their vitamin D levels (normal, deficient, or insufficient level). In bold, total number of patients based on serum category level.

Age (Years)	Insufficient	Deficient	Normal	Total
**Fractured (total):**	**216**	**301**	**171**	**688**
2–3	0	21	9	30
4–6	37	43	50	130
7–10	69	120	72	261
11–14	104	107	40	251
15–17	6	10	0	16
**Unfractured (total):**	**165**	**329**	**395**	**889**
2–3	12	48	127	187
4–6	24	67	107	198
7–10	44	81	78	203
11–14	59	86	59	204
15–17	26	47	24	97
**Total**	**381**	**630**	**566**	**1577**

## Data Availability

All data are registered at “Grigore Alexandrescu” Children’s Emergency Hospital, Bucharest, Romania.

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
