# Peer review of "Vitamin D—A Risk Factor for Bone Fractures in Children: A Population-Based Prospective Case–Control Randomized Cross-Sectional Study"

_ijerph, 2023, doi:10.3390/ijerph20043300_

Round 1

Reviewer 1 Report

This manuscript presents an evaluation of Vitamin D dosage in children/adolescent and shows their preliminary results on normal Vitamin D level (40ng/mL) in children.

The overall study design is simple, but accurate. Following few general concerns/comments and some correction.

Title: to avoid to use “population” twice in a row, I suggest to change “in the pedriatric population”, with “in children” or “during childhood”.

Line 19: Remove “the next” "Various changes occur in musculoskeletal apparatus (not body)", since in the following sentence you mentioned only bone/muscle modification.

Line 22-26: this sentence is unclear for readers. I think I have understand the core message, but, in my opinion, is poorly written. Please rephrase.

Line 36: I’m a bit confused on the “quick search on Pubmed…”. I suggest to remove the entire sentence, maybe adding few references at sentence end (line 39).

Line 42: change “certain” with “predetermined”.

Line 47: You wrote “some articles” and cited just one. Please add 1 or 2 references and give a bit of input on the sentence “to reduce health costs”.

Line 50: the sentence starts “It is estimated that..”, as a consequence you have to mention the source of this information.

Line 54: I suggest to change “there is a R in the month” and related content with “months with reduced light exposure (September-april)” or a similar sentence.

Line 56: remove “without” and use “lacking of ...”

Section from line 65 to line 77: in few lines you mentioned 3 times the same causes of Vit D deficiency. Please rewrite this section, avoiding to mention “age, sex, season, pigmentation, etc…” so many times.

Line 113: remove “recent”

Line 126: remove “coding”. ICD-10 is very famous and when you explode the acronym, you already mention the term “Classification”

Line 135: Please rewrite the sentence “were stored on digital radiography using a high-precision digital system”. Do you mean that digital imaging is stored in a repository? Or what? I do not understand.

Line 160: could you specify why you have defined the vit D and calcium subgroups (toxic, normal, poor, insufficient)? Are they related to standards or guidelines (endocrinology society)? In case, please specify them. Otherwise, please indicate a motivation to this grouping.

Statistical analysis section:  I suggest changing sentence order, indicating software analysis as last point.

Data was collected using the Institutional informatic system without influencing the medical provider. Data included categorical qualitative data (gender, type of fracture, season) and continuous quantitative data (age at diagnosis, levels of vitamin D and Calcium levels). Descriptive tests like frequencies, sex incidence, odds ratio, chi-square tests, and p-value were conducted. The results were considered significantly statistical if the p-value was less than 0.05. The statistical analysis was made by using RStudio version 1.1.447 R 3.6.0 and Microsoft Excel Office 2016. We considered a 95% confidence interval.

What do you mean with “without influencing the medical provider”? Please rephrase.

Line 184: usually a number as first word of a sentence need to be written as a word…meaning “One thousand five hundreds and seventy-seven patients…”

Line 189: please ameliorate the English…I suggest “The control group was composed of 162 (48,9%) male patients and 169 (51,05%) female patients, while the study group consisted of 237 (66,8%) boys and 120 girls (33,6%). The reference cohort included 439 males (49,8%) and 450 females (50,61%).”

Table 1: please rewrite the legend as “Patient distribution according to age grouping, gender, and group types.” The second sentence do not belong to legend. It’s a description of the results. Same for table 2. Data results must be included in the results section and not in the legends.

Line 208: add “in” before “the unfractured...”

Line 212: please indicate the exact p value (i.e. p values=0.00041), avoiding to indicate the reference value. In addition use the (dot) instead of (comma) in p values and in all the numbers (such as Vit D level) and percentages.

Line 228: modify as “(p=4.15E-08 and 0.0003 in the study group and control group, respectively)”

Lines 239-240: this sentence is part of the statistical analysis. Same for lines 278-280. You have to indicate the statistical test/approach in the statistical analysis section, while the test results in result section.

Discussion: even if well written, the discussion section is a bit long and gives too much details on ethnicity - a topic not covered by the data of the present manuscript and so not investigated. In my opinion, the authors should minimize the ethnical description of previous researches and properly discuss their results, in comparison with literature.

 In all the sections: please harmonize the term “Vitamin D” (i.e. maintain/avoid upper case) along the entire manuscript. Same for Study/study and Control/control groups (i.e. line 156, line 186, etc..)

Author Response

Thank your for reviewing our article.
Modifications were done as requested, highlited with yellow.

Title: to avoid to use “population” twice in a row, I suggest to change “in the pedriatric population”, with “in children” or “during childhood”.

RE: modification applied in line 2

Line 19: Remove “the next” "Various changes occur in musculoskeletal apparatus (not body)", since in the following sentence you mentioned only bone/muscle modification.

RE: Removed in line 19.

Line 22-26: this sentence is unclear for readers. I think I have understand the core message, but, in my opinion, is poorly written. Please rephrase.

RE: Rephrased.

Line 36: I’m a bit confused on the “quick search on Pubmed…”. I suggest to remove the entire sentence, maybe adding few references at sentence end (line 39).

RE: Removed.

Line 42: change “certain” with “predetermined”.
RE: Added in line 42

Line 47: You wrote “some articles” and cited just one. Please add 1 or 2 references and give a bit of input on the sentence “to reduce health costs”.

RE: Done.

Line 50: the sentence starts “It is estimated that..”, as a consequence you have to mention the source of this information.

RE: Done.

Line 54: I suggest to change “there is a R in the month” and related content with “months with reduced light exposure (September-april)” or a similar sentence.

RE: done

Line 56: remove “without” and use “lacking of ...”

RE: done

Section from line 65 to line 77: in few lines you mentioned 3 times the same causes of Vit D deficiency. Please rewrite this section, avoiding to mention “age, sex, season, pigmentation, etc…” so many times.

RE: Done.

Line 113: remove “recent”

RE: Done.

Line 126: remove “coding”. ICD-10 is very famous and when you explode the acronym, you already mention the term “Classification”

RE: done.

Line 135: Please rewrite the sentence “were stored on digital radiography using a high-precision digital system”. Do you mean that digital imaging is stored in a repository? Or what? I do not understand.

RE: phrase changed to "were digitally stored using a high-precision digital system.". We have a high precision digitally x-ray equipment, and all the x-ray is available on PC/mobile. We did not want to include software or hardware names.

Line 160: could you specify why you have defined the vit D and calcium subgroups (toxic, normal, poor, insufficient)? Are they related to standards or guidelines (endocrinology society)? In case, please specify them. Otherwise, please indicate a motivation to this grouping.

RE: standard laboratory values within our country. Added in lines 159-160.

Statistical analysis section:  I suggest changing sentence order, indicating software analysis as last point.

Data was collected using the Institutional informatic system without influencing the medical provider. Data included categorical qualitative data (gender, type of fracture, season) and continuous quantitative data (age at diagnosis, levels of vitamin D and Calcium levels). Descriptive tests like frequencies, sex incidence, odds ratio, chi-square tests, and p-value were conducted. The results were considered significantly statistical if the p-value was less than 0.05. The statistical analysis was made by using RStudio version 1.1.447 R 3.6.0 and Microsoft Excel Office 2016. We considered a 95% confidence interval.

Re: done

What do you mean with “without influencing the medical provider”? Please rephrase.

Re: we did not want to add comercial names. we have rephrased it.

Line 184: usually a number as first word of a sentence need to be written as a word…meaning “One thousand five hundreds and seventy-seven patients…”

Re: done!

Line 189: please ameliorate the English…I suggest “The control group was composed of 162 (48,9%) male patients and 169 (51,05%) female patients, while the study group consisted of 237 (66,8%) boys and 120 girls (33,6%). The reference cohort included 439 males (49,8%) and 450 females (50,61%).”

RE: done!

Table 1: please rewrite the legend as “Patient distribution according to age grouping, gender, and group types.” The second sentence do not belong to legend. It’s a description of the results. Same for table 2. Data results must be included in the results section and not in the legends.

RE: done!

Line 208: add “in” before “the unfractured...”

RE: done

Line 212: please indicate the exact p value (i.e. p values=0.00041), avoiding to indicate the reference value. In addition use the (dot) instead of (comma) in p values and in all the numbers (such as Vit D level) and percentages.

RE: Done.

Line 228: modify as “(p=4.15E-08 and 0.0003 in the study group and control group, respectively)”

Re: done!

Lines 239-240: this sentence is part of the statistical analysis. Same for lines 278-280. You have to indicate the statistical test/approach in the statistical analysis section, while the test results in result section.

RE: lines removed

Discussion: even if well written, the discussion section is a bit long and gives too much details on ethnicity - a topic not covered by the data of the present manuscript and so not investigated. In my opinion, the authors should minimize the ethnical description of previous researches and properly discuss their results, in comparison with literature.

RE: We elaborated as much as we found on every aspect of Vitamin D deficiency. We considered it to be useful information for future researches. Skin color can alter therapeutic dosages of Vitamin D. 

 In all the sections: please harmonize the term “Vitamin D” (i.e. maintain/avoid upper case) along the entire manuscript. Same for Study/study and Control/control groups (i.e. line 156, line 186, etc..)

RE: Done.

Reviewer 2 Report

This is an interesting study about the risk of bone fractures in children with low levels of vitamin D. Having a bone fracture during infancy and the disease associated with low vitamin D levels are scientifically both hot topics of discussion. This study is a prospective study evaluation this topic and the number of children evaluated is very high. However, I think on the whole the article should be thoroughly revised before publication:

Introduction:

-The risk of fractures on children with low levels of vitamin D is not clearly stated in the introduction, but after in the discussion this is stated and referenced more than once. I think the introduction should be a state of the art about the topic, and the Discussion, a discussion about the results of the study.

-Lines 87-88: the reference is a review article, not the actual recommendations of the AAP.

-Consider including the more actual consensus recommendations for vitamin D:  Munns CF, Shaw N, Kiely M, et al. Global Consensus Recommendations on Prevention and Management of Nutritional Rickets. J Clin Endocrinol Metab. 2016;101(2):394-415. doi:10.1210/jc.2015-2175

Results:

-Line 295-299: it is hypothesised in the Introduction that the healing process is delayed in children with lower levels of 25OH-vitamin. However, there are no objective data showing this hypothesis in the results. I suggest either showing objective data no support this hypothesis or removing it.

Discussion: Half of the discussion does not discuss the results but is a continuation of the introduction (see commentary in the Introduction).

The statement “ Our preliminary results show that the normal level of Vitamin D in children should start at 40ng/mL” should be completed. Data of your study at the most show that this could be a correct level to prevent some fractures in children.

Some references are used arbitrarily, for example, lines 312-314 do not correspond with reference 30 (it is not the recommendation of the Endocrine society). Lines 307-311, reference 29 does not speak about refractory rickets in the tropics but is a review.

Author Response

Thank you for reviewing our article.
Modifications were done as requested, highlighted with yellow.

Introduction:

-The risk of fractures in children with low levels of vitamin D is not clearly stated in the introduction, but after the discussion this is stated and referenced more than once. I think the introduction should be state of the art about the topic, and the Discussion, a discussion about the results of the study.

RE: Introduction contains general information about the role of vitamin D, the means to modulate serum level of vitamin D, and our hypothesis. We did not know the risk from our study pool before statistical analysis. We chose to compare our results with the literature and to elaborate on others' findings, as is usually stated in the Discussion. 

-Lines 87-88: the reference is a review article, not the actual recommendations of the AAP.

RE: The paper we cited contained the recommendations that the AAP endorses (Table 2 in the form). Was removed.

-Consider including the more actual consensus recommendations for vitamin D:  Munns CF, Shaw N, Kiely M, et al. Global Consensus Recommendations on Prevention and Management of Nutritional Rickets. J Clin Endocrinol Metab. 2016;101(2):394-415. doi:10.1210/jc.2015-2175

RE: Added on lines 83-85.

Results:

-Line 295-299: it is hypothesized in the Introduction that the healing process is delayed in children with lower levels of 25OH-vitamin. However, no objective data are showing this hypothesis in the results. I suggest either showing objective data not support this hypothesis or removing it.

RE: In our study, we noticed that the healing process is delayed in children with lower levels of 25-OH-Vitamin D, thus we showed an example in Figure 2 line 282, and explained in lines 289-290. We had the patients compared at the same time in evolution (3 weeks, 5 weeks, etc.), same fracture, age-sex matched, same lateral and AP x-rays, and we noticed that bony callus was even more present for the patients that received vitamin D and Calcium.

Discussion: Half of the discussion does not discuss the results but is a continuation of the introduction (see commentary in the Introduction).

RE: Indeed we elaborate a lot in our Discussion. We tried to mention the papers that agree with our results and also the papers that do not, and the reasons their results were different (fracture location, ethnicity, age). Discussions were expanded in lines 301-302, 311-314, 319-323, 331-334, and 338-340.

The statement “ Our preliminary results show that the normal level of Vitamin D in children should start at 40ng/mL” should be completed. Data from your study at the most show that this could be the correct level to prevent some fractures in children.

RE: rephrased in lines 381-382

Some references are used arbitrarily, for example, lines 312-314 do not correspond with reference 30 (it is not the recommendation of the Endocrine society). Lines 307-311, reference 29 does not speak about refractory rickets in the tropics but is a review.

RE: We made the modifications. reference 29 is "Sun exposure as a strategy for acquiring vitamin D in developing countries of the tropical region: Challenges & way forward"

Round 2

Reviewer 2 Report

I thank the authors for their answers to my comments. Some corrections and comments to the new version of the manuscript:

1.  I think the references should be revised:

-Lines 82-84: reference is incorrect (not 22 but 21).

-Lines 366-369: still refers to the American Academy of Pediatrics, reference 21, but this is not the case. Reference 21 is a global consensus recommendation

2. I insist that the results about the healing process delayed in children with lower vitamin D (lines 289-292) are interesting but should be more developed (as you answered: “. We had the patients compared at the same time in evolution (3 weeks, 5 weeks, etc.), same fracture, age-sex matched, same lateral and AP x-rays, and we noticed that bony callus was even more present for the patients that received vitamin D and Calcium”.

This is an interesting finding and it is a part of the hypothesis of your work. However, you show lots of tables and results about the levels of vit D and the risk of fractures in the different groups, but nothing about the healing process. I think this should be developed in the Result section and the Discussion.

Author Response

Dear Reviewer,

Thank you for your careful assessment.

We made the following modifications:

 I think the references should be revised:

-Lines 82-84: reference is incorrect (not 22 but 21).

A:Done.

-Lines 366-369: still refers to the American Academy of Pediatrics, reference 21, but this is not the case. Reference 21 is a global consensus recommendation

A:Done.

I insist that the results about the healing process delayed in children with lower vitamin D (lines 289-292) are interesting but should be more developed (as you answered: “. We had the patients compared at the same time in evolution (3 weeks, 5 weeks, etc.), same fracture, age-sex matched, same lateral and AP x-rays, and we noticed that bony callus was even more present for the patients that received vitamin D and Calcium”.

A: We completed our Results with the healing time in lines 288-294 and we mentioned the only article we found about healing time among children who took Vitamin D supplements (a case presentation) in lines 375-381. We did notice improvement, which could be useful in further research.
